electromagnetism/materials science/ electrical engineering

additive manufacturing, wireless communications, orbital angular momentum, spiral phase plate

**Author for correspondence:**
D. Isakov
e-mail: d.isakov@warwick.ac.uk

# Evaluation of the Laguerre–Gaussian mode purity produced by three-dimensional-printed microwave spiral phase plates

D. Isakov[1,2], Y. Wu[2], B. Allen[3], P. S. Grant[2], C. J. Stevens[3] and G. J. Gibbons[1]

[1]WMG, University of Warwick, Coventry CV4 7AL, UK
[2]Department of Materials, University of Oxford, Oxford OX1 3PH, UK
[3]Department of Engineering Science, University of Oxford, Parks Road, Oxford OX1 3PJ, UK

  DI, 0000-0002-3913-8096; YW, 0000-0003-1334-6193;
BA, 0000-0002-6308-8383; PSG, 0000-0002-7942-7837;
GJG, 0000-0001-8722-6165

Computer-aided design software and additive manufacturing provide flexibility for the direct fabrication of multi-material devices. This design and fabrication versatility has been investigated for the manufacture of dielectric spiral phase plates (SPP) that generate electromagnetic waves with helical wavefronts. Three types of SPPs designed to produce an orbital angular momentum (OAM) mode number $l = |1|$ were additively manufactured using material extrusion and polyjet fabrication methods. The OAM mode characteristics of the transformed helical microwaves as a function of the SPP geometrical features were investigated experimentally in the 12–18 GHz frequency range. The SPPs were further combined with an additively manufactured dielectric lens that provided a marked improvement in OAM mode purity. Finally, multiplexing and de-multiplexing of two OAM modes were demonstrated successfully using an optimum SPP geometry and arrangement.

## 1. Introduction

Spiral phase plates (SPPs) are optical devices that transform a plane electromagnetic wave to a Laguerre–Gaussian (LG) beam. The LG beam then carries orbital angular momentum (OAM) with a non-zero LG mode number that is characterized by a helical phase front with the axis of rotation along the boresight, or direction of propagation. The screw discontinuity of the phase pattern results

in a doughnut-like profile of the magnitude of the beam, with a null magnitude along the boresight [1,2]. The helicoid shape of the wavefront has a step $\lambda/l$, where $\lambda$ is the wavelength and $l = \pm 1, \ \pm 2, \ \pm 3$ determines the rotational mode (topological charge of the phase vortex) with an azimuthal structure related to $\exp(il\varphi)$, where $\varphi$ is the azimuthal coordinate [1,3].

The general principle for the generation of a helical wave with OAM is an induced azimuthal phase delay proportional to $\exp(il\varphi)$ in the Gaussian beam. Typically, OAM can be obtained using a spatial light modulator or forked diffraction grating generating an interference pattern hologram that converts a plane wave into a beam with a pre-defined phase and amplitude structure [1]. The helical wavefront of an LG beam makes coaxial beams mutually orthogonal, therefore in principle minimizing any inter-mode cross-talk and allowing the use of multiple OAM modes simultaneously. OAM modes can be used to enhance the security of a transmission channel [4] and spectral efficiency [5].

There is much discussion in the literature on the application of LG-type beams carrying OAM to multiplex systems to increase data transfer capacity [4,6,7] and OAM mode division multiplexing has been shown to be promising for high-capacity wireless communication lines [8,9], although contrasting opinions also exist [10–12]. It is fair to note, however, that OAM communication provides an optimal solution only under very specific conditions.

In the microwave frequency range, for the generation of LG modes, a variety of methods have been proposed. The most common is an antenna array (typically short dipoles or patches) arranged around a circumference and producing a signal with an incremental phase shift sequentially over the array elements [4,13]. Generation of OAM mode $l = 3$ has been demonstrated at 18 GHz using a Cassegrain antenna fed by a $4 \times 4$ waveguide Butler matrix [14]. Instead of expensive phase shifters, a time-modulated circular array, where individual antenna elements were sequentially switched on/off has also been used to realize OAM beams [15]. Such an implementation allows the generation of multiple OAM-carrying beams at several harmonic frequencies simultaneously, as well as enhancing the field intensity of each mode [16]. A microstrip ring with a matched load (phase changes along a circle of $2\pi l$) has been proposed to produce a circular travelling wave with a double OAM mode at 5 GHz [17]. Radiating element design is especially convenient for integration into radio frequency circuits. Another design for a dual OAM mode antenna has been proposed by Hui *et al.* [18] based on a circular slot cavity resonator where the resonator is a metallic ring with two rectangular feed ports operating with a $\pm\pi/2$ phase difference to excite clockwise or anti-clockwise travelling wave distributed fields.

Multiple designs of metasurfaces (composed of subwavelength dipoles or aperture 'meta-atoms' arrays with specifically pre-defined shapes, orientation and location) have also been used widely for the transformation of a plane wave into an LG mode. A meta-axicon composed of concentrically arranged plasmonic nano-apertures can provide phase shifts for the transmitted light [19]. An anisotropic frequency selective surface with anisotropic responses in orthogonal directions made of cross-dipole elements [20,21], split square resonators [22,23], layered square patches with a cross central slot [24], dielectric nanofins [25] have all been shown to convert a linearly polarized wave from a single point source into an LG beam. A plane radio wavefront can also be twisted using a properly shaped surface, e.g. a hemispherical dielectric resonator antenna [26] or a parabolic helical reflector [27].

Alternatively, the LG mode can be induced by a dielectric plate with an azimuthal-dependent optical thickness of $tn = (\lambda/2\pi)l\varphi$, where $t$ is the thickness and $n$ is the refractive index of the plate. The emitted helical phase can be controlled by either a flat plate with azimuthal-varying permittivity $n^2$ [28,29] or a plate with azimuthal-varying thickness [30,31]. The key point is that in contrast to a metamaterial SPP or a reflecting surface, a dielectric SPP can produce a phase vortex with any polarization.

In this paper, we exploit additive manufacture (AM, also known as three-dimensional (3D) printing, 3DP) to realize dielectric SPPs. We investigate how different designs of the SPP can affect the quality of the phase transformation of a plane wave into a wave with a helical front. The design and implementation freedom provided by inter-linked computer aided design (CAD) and AM methods made the fabrication of arbitrary geometrically complex structures relatively easy. Access to various methods of AM allowed a sufficient surface finish and tolerance to be achieved. Three SPPs were investigated: (i) an SPP with a smooth surface and azimuthally increasing thickness made of relatively low permittivity acrylic resin; (ii) a 'staircase' SPP with the stair thickness azimuthally increasing made of a composite of relatively high dielectric permittivity, and (iii) a flat SPP with alternating regions of relatively high and low permittivity material the proportions of which varied with the azimuthal position. The SPPs were intended to provide essentially similar overall geometry and phase transformation, but with different arrangements of the boundaries between materials of varying refractive index, which might provide for different internal reflection behaviour and output transmission functions carrying the OAM. We then explore the use of the SPP in a simple multiplexed communication system.

# 2. Methods

The SPP, horn antennas, and lenses used in the measurement campaign were a bespoke design and fabricated by AM. The horns had a rectangular cross-section and designed to achieve a high directivity of 18 dBi at 15 GHz. The horns were fed by commercial Ku-band SMA coax via waveguide adaptors (Flann Microwave) connected to a vector network analyser (Rhode&Schwarz ZNB20) for excitation and response measurement in the frequency range 12–18 GHz. A computer-controlled $X$–$Y$ gantry system was used to move the receiver horn in a pre-programmed pattern to map complex scattering parameters in the plane perpendicular to the beam direction, positioned at approximately $35\lambda$ from the stationary source horn.

SPPs with staircase and flat gradient refractive index (GRIN) designs and with $l = \pm1$ were 3D-printed using a fused filament fabrication dual-head desktop 3D printer (Makerbot 2X) using bespoke acrylobutene styrene (ABS)/particulate BaTiO$_3$ feedstock filaments with relatively high dielectric permittivity of $\approx$11, tan $\delta$ = 3.03 × $10^{-2}$ [32] and commercial ABS with relatively low dielectric permittivity = 2.65, tan $\delta$ = 4.69 × $10^{-3}$ [33]. The SPP with continuous azimuthally increasing thickness was manufactured using a PolyJet 3D printer (J750, Stratasys) using commercial acrylic photopolymer (VeroClear, Stratasys) with a dielectric relative permittivity = 3.00 ± 0.05, tan $\delta$ = 6.8 × $10^{-3}$ at 15 GHz. The geometry of each SPP was designed based on calculations using the measured dielectric properties of the different materials previously determined using a split-post resonator technique (QWED, Poland) at a nominal frequency of 15 GHz. The size of the SPPs was chosen to fully cover the aperture of the antenna horn.

A dielectric bi-convex lens was used to collimate the divergence of the OAM waves and was manufactured using a PolyJet 3D printer (J750, Stratasys). The lens was 90 mm in diameter and had a focal length of 110 mm at 15 GHz, and further details can be found in [34].

# 3. Results and discussion

## 3.1. Characteristics of a beam with helical wavefront

Soskin et al. [35] showed that a wave equation solution for a Gaussian envelope carrying an optical vortex with charge $l$ is an eigenmode of the paraxial Helmholtz equation:

$$E(\rho, \theta, z) = E_0 \frac{w_0}{w} (\frac{\rho}{w})^{|l|} \exp\left(-\frac{\rho^2}{w^2}\right) \exp\left[i\Psi(\rho, \theta, z)\right], \quad (3.1)$$

where $E_0$ is the amplitude, $w_0$ and $w$ are the waist radius and beam width respectively. The phase $\Psi$ is given by

$$\Psi(\rho, \theta, z) = -(|l| + 1)\arctan\frac{z}{z_R} + \frac{k\rho^2}{2R(z)} + l\theta + kz, \quad (3.2)$$

where $z_R$ is the Rayleigh length, $R(z)$ is the radius of the wavefront curvature, and $k$ is the wavelength. This solution is, in essence, a $LG_0^l$ mode. The equi-phase surface $z$ with a radius of curvature $R(z)$ at a fixed elevation angle $\theta$ defines a helix twisted along $z$-axis with a pitch of $l\lambda$, where $\lambda$ is wavelength [36].

## 3.2. Spiral phase plate design and fabrication

Figure 1 presents the CAD drawings and corresponding images of the three 3D-printed SPPs. For the SPP with a smooth surface (figure 1a), the thickness of the plate continuously increased with azimuthal angle reaching a maximum thickness of $th = 27.3$ mm. The staircase design of the SPP is shown in figure 1b where the azimuthal dependence of the thickness was changed discretely in increments of step $th_s = \lambda/(n-1)\alpha/2\pi$, where $n$, $\lambda$ and $\alpha$ is the refractive index, wavelength and incremental angle of the step respectively. Using an in-house bespoke filament with permittivity approximately 11 the total thickness of the SPP was reduced from 27.3 mm to 8.6 mm. To partially compensate for the reflection loss caused by the large difference in index of refraction when using this high permittivity material, the SPP was printed with an index matching layer (IML) made of ABS with a lower relative permittivity of 2.65 [33] and a thickness of 5 mm.

The final SPP in figure 1c comprised the gradient index design where the azimuthal variation of the optical path was changed by means of a local change in the effective refractive index, rather than by thickness changes. The flat SPP design was split into 28 wedges of variable size—14 comprised high

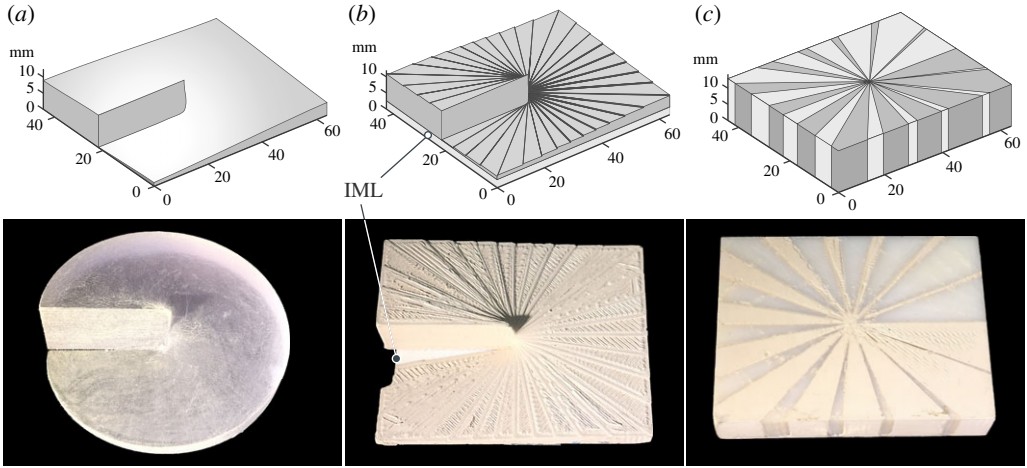

**Figure 1.** CAD models and corresponding photographs of the 3D-printed dielectric spiral phase plates: (*a*) with smoothly varying thickness, acrylic polymer $\varepsilon = 3 - j0.020$; (*b*) with a staircase structure, ABS/BaTiO$_3$ composite, $\varepsilon = 11 - j0.330$; and (*c*) flat with gradient refractive index, ABS/BaTiO$_3$ composite and pure ABS with $\varepsilon = 2.60 - j0.012$.

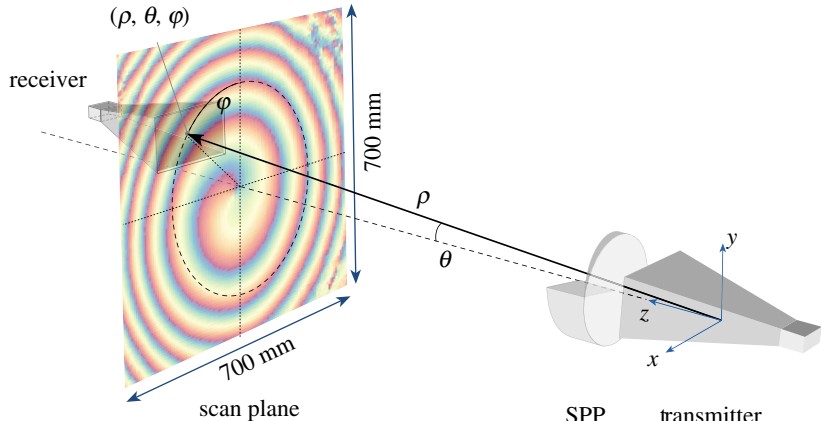

**Figure 2.** The experimental arrangement for the measurement of microwaves with OAM induced by a dielectric 3D-printed spiral phase plate (SPP) placed in front of the transmission horn.

permittivity BaTiO$_3$/ABS and 14 comprised low permittivity ABS. 'High/low' pairs of wedges formed 14 equal-sized segments in which the relative proportion (or in-plane area) of the different wedges changed according to azimuthal position (figure 1*c*) to produce a different average, effective permittivity, simply given by the volume-weighted proportion of the two sub-wedges. The 14 segments (total 28 wedges) were the smallest discrete segments that could be reliably printable (further increasing the wedge number and decreasing the wedge size made the required resolution in the central part of the SPP too fine-scale to be printed). By keeping the thickness of the plate constant at $\lambda/\Delta n = 11.84$ mm, the area ratio of the high and low permittivity was changed with azimuthal rotation to produce a distribution of refractive index such that the maximum generated phase delay for a wave with $\lambda = 19.98$ mm (15 GHz) was $2\pi$ radians.

## 3.3. Analysis of the mode purity

The performance of the SPPs and their ability to generate a beam with induced OAM were characterized by analysis of the distribution of the transmitted signal. The SPP was placed in front of a horn aperture and the transmitted $S_{21}$ complex signal was recorded over the range 12–18 GHz as previously described. The scanning area was $700 \times 700$ mm with a step size in the range 2–5 mm, as shown in figure 2.

As an example, figure 3 shows the typical amplitude and phase of the signal generated in the *X*–*Y* plane perpendicular to the direction of propagation, measured with 5 mm step resolution. The amplitude pattern (top row of images in figure 3) exhibited a doughnut-like beam profile with null

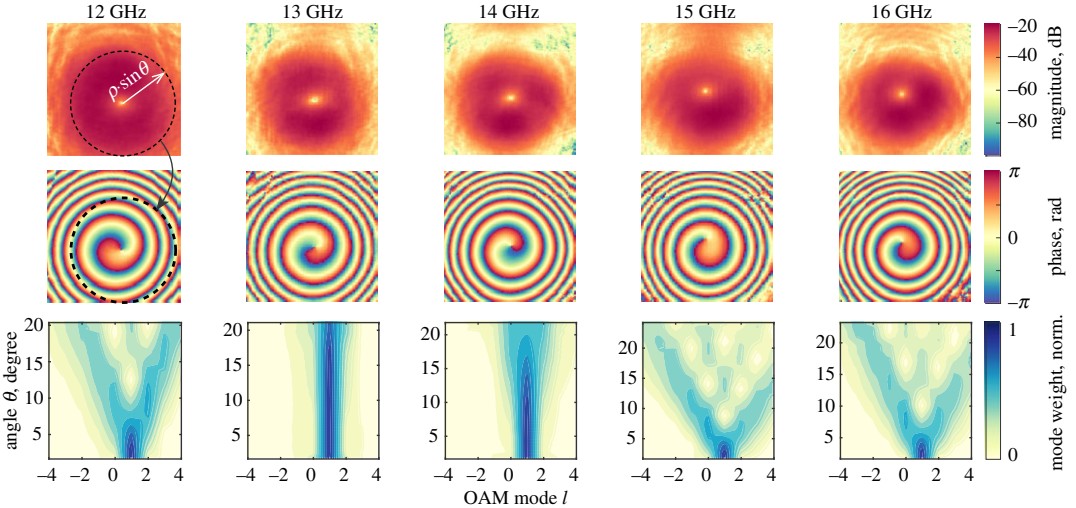

**Figure 3.** The radiation signal passed through an SPP with a staircase design (as an example) in an X–Y plane of area 700 × 700 mm² perpendicular to the propagation vector. The experimental arrangement is shown in figure 2. Top row—the $S_{21}$ amplitude profile; middle row—the $S_{21}$ phase profile; bottom row—the OAM mode spectrum. The circle superimposed on the top row, left corner figure has its centre at the location corresponding to the minimum of the amplitude intensity. The corresponding phase data on the second row was obtained by moving along the data points on the circle circumference and at each point decomposing the amplitude data using a Fourier transform to obtain the phase profile (second row) and the OAM mode (third row).

intensity in the centre, whereas the phase profile (middle row in figure 3) exhibited the expected vortex pattern for OAM of mode $l = +1$.

The OAM mode purity was determined using mode spectrum analysis [22]. For a beam with a phase vortex, the field in the plane perpendicular to the beam direction is given by

$$E(\rho, \varphi, \theta) \propto A(\rho) \cdot \Psi(\varphi, \theta) \cdot e^{il\varphi}, \tag{3.3}$$

where $A$ is the amplitude and $\Psi$ is the phase corresponding to the point $(\rho, \varphi, \theta)$. Thus, the azimuthal phase dependency is carrying the information about phase helicity with mode $l$. Owing to the $2\pi$ periodicity of the angular position of the beam, the OAM spectrum can be obtained by mode decomposition using a Fourier transform [37]. The Fourier relationship between the distribution of angular momenta $\beta$ and the angular distribution $\Psi(\varphi)$ is given by [37,38]

$$\beta(l) = \frac{1}{\sqrt{2\pi}} \int_0^{2\pi} \exp(-jl\varphi) \Psi(\varphi) \, d\varphi, \tag{3.4}$$

where $\varphi$ is the azimuthal angle, and $\Psi(\varphi)$ is the angular distribution of the phase observed along the circumference with radius $\rho \sin\theta$.

As an example, the OAM mode spectrum as a function of elevation angle $\theta$ (see figure 2 for reference) is shown in the bottom row of images in figure 3 for the SPP with a staircase structure and for frequencies in the range 12–16 GHz. The corresponding amplitude, phase and mode spectra for the other 3D-printed SPPs are shown in the electronic supplementary material, figure S1. The colour intensity key indicates the weight of the OAM mode (mode power ratio) $P_l$, defined as the fraction of power of the fundamental mode to all modes:

$$P_l = \frac{\int \int |E_l(\varphi, \theta)|^2 \sin\theta \, d\theta d\varphi}{\sum_{m=-\infty}^{+\infty} \int \int |E_{l+m}(\varphi, \theta)|^2 \sin\theta d\theta d\varphi}, \tag{3.5}$$

where the index of the electric field corresponds to the OAM mode, with $l$ as the intended mode and $l + m$ representing mixed modes carried by the vortical wave. The mode energy ratio can be considered as a figure of merit for the ability of the SPP to generate OAM.

As shown in figure 3, the mode purity spectrum plot varied with frequency with the contribution of undesired modes expressed as divergent 'branches' forming a 'V' shape in the OAM spectra on the bottom row. For this case of the 3D-printed SPP with a staircase structure, the signal at 13 GHz had the highest purity across the elevation angle range, up to 22° (physically limited by the scanning area and distance from the source to the scanning plane of the equipment). The poorest purity of the OAM

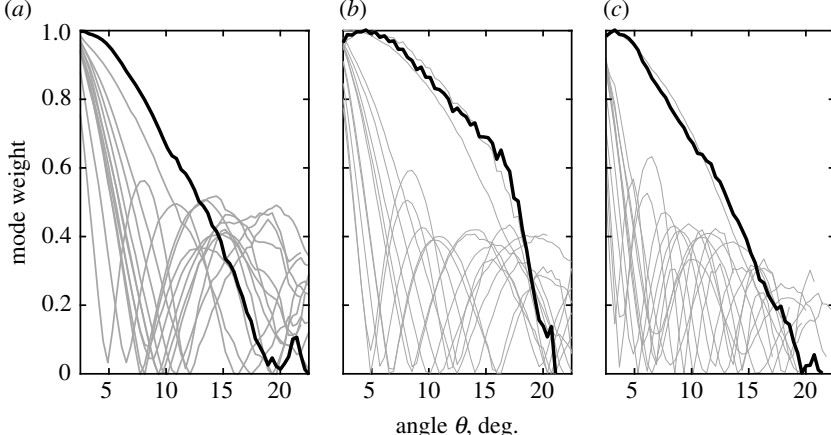

**Figure 4.** Mode energy ratio as a function of elevation angle for the 3D-printed SPPs. Each grey plot-line represents a different discrete frequency. (*a*) SPP with a smooth curvature surface; (*b*) SPP with a staircase structure; and (*c*) a flat SPP with the gradient refractive index. The bold lines in each subfigure represent the frequency at which OAM mode had the highest overall purity.

mode $l = 1$ was at 15 GHz, the frequency for which the SPP was designed and had been expected to show optimal performance. This was owing to the imperfection during the fabrication (such as air voids between the polymer toolpath, typical for extrusion material 3DP) resulting in overall decrease in effective dielectric permittivity of the SPP.

Figure 4 shows the intended mode ($l = |1|$) weight (also referred to as the mode energy ratio) for all three printed SPPs as a function of the elevation angle $\theta$, in the 12–18 GHz range in 0.5 GHz steps. Each of the lines is the experimental data for a specific frequency, as the angle was varied over the range. In most cases, the intended mode could only be maintained at relatively small elevation angles, typically $\theta < 5°$ beyond which the mode degenerated owing to overlapping of multiple phase modes. The thicker black line in each figure indicates the frequency that provided the overall best performing, purest response (over the range) for each SPP. The detailed behaviour of the mode weight depended on the geometry of the SPP. For example, for the SPP with a smooth curvature surface (figure 4*a*) and flat SPP (figure 4*c*), the intended mode response decayed rapidly with increasing angle. The secondary peaks superimposed on the decay corresponded to the V-branches of the parasitic modes previously described for the OAM mode spectra on the bottom row of figure 3. Among the three SPPs, the SPP with the staircase structure in figure 4*b* had the most favourable response.

Taken together, the results in figures 3 and 4 were in good qualitative agreement with the theoretical analysis described in [38,39] for a vertex beam generated by an antenna array, which also showed that the energy of the fundamental mode decreased with increasing elevation angle, whereas the energy of the harmonic modes increased. Whereas a previous comparison between SPPs with either stepped or smooth azimuthal thickness changes suggested a small advantage for the smooth case [31], our results for modes $l = \pm1$ show that stepped SPP had a small percentage improvement over the smooth SPP. However, in both cases, the differences were relatively small.

Minimizing inter-modal interference is a key parameter in the design of an OAM link. As with any radio link, sufficient signal-to-interference-plus-noise ratio would have to be achieved to restrict the bit error rate. A further particular aspect of an OAM-based communication system is the possibility that it may offer for enhanced security, and this is the focus of current work.

## 3.4. Reducing orbital angular momentum beam divergence

From the OAM modal spectrum and mode power ratio, a critical elevation angle $\theta_c$ (purity threshold) can be evaluated. Here, we define the critical angle $\theta_c$ as the angle at which the power carried by the fundamental azimuthal component was 75% of the total power. This means that the region bounded by this angle represents the region with high purity of the intended OAM mode $l$. Figure 5 shows the critical elevation angle for all three SPPs as a function of frequency. The 3D-printed SPP with a smooth surface (circles in figure 5*a*) showed weak mode purity with $\theta_c < 5°$, apart from a peak at approximately 10° and 16 GHz.

Similar but better performing (higher $\theta_c$) results are also shown for the staircase and GRIN structure SPPs in figures 5*b*,*c* respectively. As already stated, among the three SPPs, the staircase structure

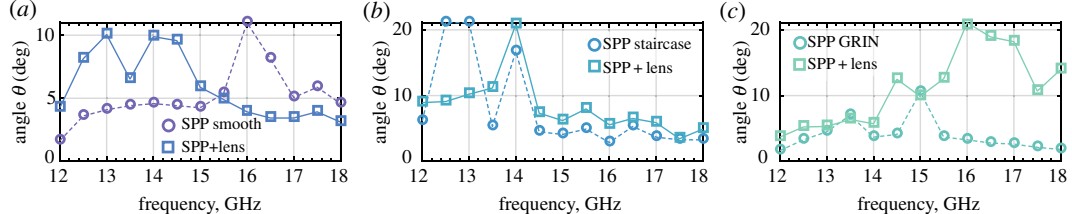

**Figure 5.** The threshold critical angle as a function of frequency for (*a*) SPP with a smooth curvature surface; (*b*) SPP with a staircase structure; and (*c*) flat SPP with gradient refractive index.

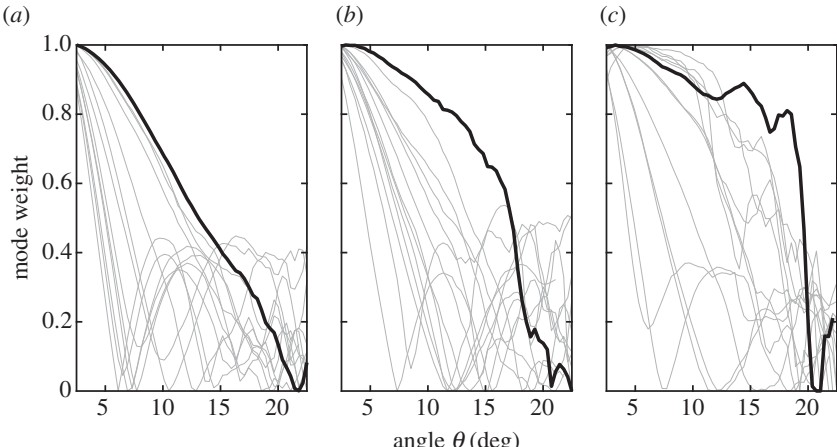

**Figure 6.** Mode energy ratio as a function of evaluation angle $\theta$ for a microwave link system composed of a 3D-printed SPP and 3D-printed lens. (*a*) SPP with a smooth curvature surface; (*b*) SPP with a staircase structure; and (*c*) a flat SPP with gradient refractive index. The bold lines correspond to the frequency at which the OAM mode is the purest.

(figure 1*b*) showed the best overall performance regarding mode purity, generating the OAM $l = +1$ mode across the whole scanning area at 12.5–14 GHz.

Previous work by Allen *et al.* [40] revealed that by using a dielectric 3D graded index lens, the beam divergence could be reduced that in turn allowed a doubling of the OAM radio link distance between an antenna array generating a vortical beam and a receiver. Therefore, to reduce beam divergence and to investigate the effect of lensing on OAM mode purity, a dielectric bi-convex lens with focal length 110 mm and a diameter 90 mm was 3D-printed using a photocurable resin of refractive index 1.73 [34]. The lens was placed directly after the SPP at a distance from the transmission horn aperture that was the focal length of the lens.

The critical elevation angle $\theta_c$ for the link system with the lens is shown in figure 5 (squares) for all three SPPs. The lens significantly increased the area with acceptable mode purity. Also, the dielectric lens exhibited an additional gain of 11 dBi. Therefore, using a dielectric lens had two positive effects on the OAM beam: first, it reduced beam divergence and thereby potentially increasing the useful distance of the communication transmission; and second, it improved the purity of the helical mode generated by the 3D-printed SPPs. Moreover, in certain cases, the use of a dielectric lens significantly increased the purity threshold, i.e. the area of the radio beam carrying the desired mode with maximum energy.

Figure 6 presents the mode energy ratio (mode weight) for the set of 3D-printed SPPs as a function of evaluation angle with the dielectric lens. There was a significant change in the performance of the GRIN SPP when the lens was used (figure 6*c*). Together with an increase of the critical elevation angle $\theta_c$ for mode $l = +1$, at higher frequencies (15–18 GHz) the mode weight curve had a step-like shape (bold curve) with a high mode purity within the area defined by $\pi\rho 2\sin\theta_c$, where $\theta_c = 19°$. With the lens, the GRIN SPP was now the best performing.

## 3.5. Orbital angular momentum mode multiplexing

A further experiment was implemented to investigate the use of the SPP in multiplexing the $l_2 = 1 + 1$ modes, and/or de-multiplexing $l_0 = 1 - 1$ mode mixing [34]. A schematic of the experimental

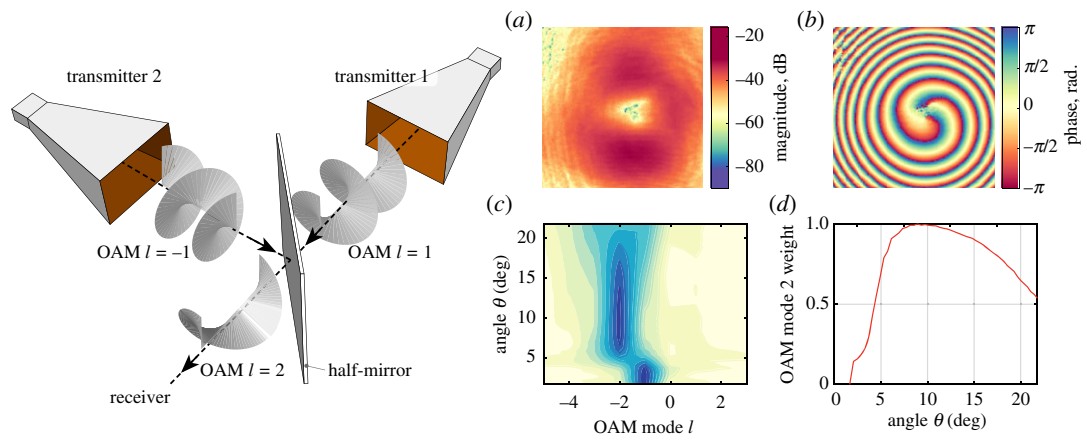

**Figure 7.** Multiplexing of two OAM modes with $l = 1 + 1$. Mode $l = -1$ changed its polarity owing to the reflection. Left insert: a schematic of the experimental set-up. When the half-mirror was oriented at $\pi/4$ to the incident beam, 50% of the incident beam intensity was refracted and 50% was transmitted. The resulting (a) $S_{21}$ amplitude profile; (b) $S_{21}$ phase profile; (c) OAM mode spectrum; and (d) mode energy ratio.

arrangement is shown in the inset in figure 7. Two transmitting horns were attached to SPPs (not shown) that were perpendicularly oriented to one another. At the virtual beam geometrical crossing point, an $Al_2O_3$ square plate of area 100 cm² and thickness 2 mm was placed to act as a semi-transparent microwave mirror. The mirror was rotated $\pi/4$ radians to both beam directions about its vertical axis. Thus, if the required accuracy in the alignment of all elements could be achieved, both transmitted and reflected OAM beams should remain collinear. To obtain a summation of OAM modes, SPPs with positive and negative topology must be used simultaneously, owing to the phase sign change on reflection. This type of 90° orientation of the horns for mode multiplexing is preferable to a linear, in-line sequence of two SPPs one after another that tends to suffer from higher loss.

Figure 7a,b shows the amplitude and phase profiles at 13 GHz of the multiplexed, mixed signals passed through staircase SPPs with $l = 1$ and $l = -1$ modes. Compared with single mode results (figure 3), the amplitude had a more distinct null node in the centre while the phase profile had a pronounced double-helix depicting the $l = -2$ mode. The negative sign of the resulting OAM mode could be changed simply by swapping the SPPs.

Figure 7c shows the corresponding mode spectrum, with a predominant OAM mode $l = -2$ in the elevation range 4°–20°. The secondary mode $l = -1$ was resolved only at a relatively small angle in the vicinity of the phase discontinuity. The mixed signal had a high mode $l = -2$ purity. Overall, the mode purity was superior to that of a single OAM beam (a wider area of higher mode power ratio). This improvement can be explained by the expansion of the LG beams with OAM modes $l \neq 2$. An implication of these results is that if two SPPs of identical topological charge were used, a beam with OAM mode $l = 0$ (a plane wave), would result. In other words, a plane wave at source could be transmitted as an OAM beam, and the plane wave recovered on receipt. Successful results for this demodulation $l = 1 - 1$ using the arrangement are shown in the electronic supplementary material.

## 4. Conclusion

In order to use microwave waves carrying OAM in practical communication systems it is important that the OAM mode has high purity to allow efficient use of the transmission line without energy leakage to other modes. The main problem in achieving this in practice is cross-interference between OAM modes. In this work, we have analysed the OAM mode purity of signals generated by three different dielectric SPPs. Three SPPs with a designed overall topological charge $l = |1|$ used different arrangements of materials and geometry, and were fabricated by multi-material 3D-printing. The different SPP geometries gave different microwave beam refraction characteristics. SPPs with both free surfaces perpendicular to the incident beam direction—a SPP with an azimuthal staircase structure and a parallel sided SPP with an azimuthal gradient refraction index—tended to provide OAM modes of relatively high, and useful, purity. Surprisingly, the third SPP that had a smooth azimuthal variation in thickness of a single material showed the lowest overall OAM mode purity, because: (i) the 'upper'

(figure 1a) non-normal free surface promoted internal reflections that contributed to interference and superposition of radial modes; and (ii) the discontinuity produced by the single large step was of thickness close to the incident wavelength and may have promoted a tendency for internal standing waves. Combining the 3D-printed SPPs with a 3D-printed dielectric lens produced distinct improvements in OAM mode purity and reduced OAM beam divergence. The best overall OAM purity and reduced OAM beam divergence was generated by a flat, relatively thin SPP with an azimuthally varying graded index combined with a dielectric lens. Subsequently, two SPPs and a dielectric half-mirror were then assembled into a rudimentary microwave multi-mode multiplexed transmission and reception system. Preliminary data successfully confirmed multi-mode and multiplexed transmission and reception for a range of SPP configurations.

Data accessibility. This article has no additional data.

Authors' contributions. D.I. designed the SPPs, managed all the experiments and drafted the manuscript, Y.W. performed experiments and contributed to the manuscript, B.A. and C.J.S. contributed to the analyses and interpretation, P.S.G. and G.J.G. directed the research and proofread the manuscript.

Competing interests. We declare we have no competing interests.

Funding. The authors thank the UK Engineering and Physical Science Research Council for financial support (grant no. EP/P005578/1).

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
