## [Reviewer comments · Royal Society Open Science]

Review History

RSOS-200493.R0 (Original submission)

Review form: Reviewer 1

Is the manuscript scientifically sound in its present form?

Yes

Are the interpretations and conclusions justified by the results?

Yes

Is the language acceptable?

Yes

Do you have any ethical concerns with this paper?

No

Have you any concerns about statistical analyses in this paper?

No

Recommendation?

Major revision is needed (please make suggestions in comments)

Comments to the Author(s)

The manuscript presents the design and implementation of spiral phase plates (SPPs) using additive manufacturing for generating OAM waves. Below are my comments and questions on the presented work:

-Based on the introduction, the main contribution of the presented work is not clear. Please add explanations clarifying the main contributions behind the presented research work specially in comparison with the following article, which has been recently published by the same authors.

Allen B, Pelham T, Wu Y, Drysdale T, Isakov D, Gamlath C, Stevens C, Hilton G, Beach M, Grant PS. 2019 Experimental evaluation of 3D printed spiral phase plates for enabling an orbital angular momentum multiplexed radio system, B. Allen, T. Pelham, Y. Wu, T. Drysdale, D. Isakov, C. Gamlath, C. Stevens, G. Hilton, M. Beach and P.S Grant, Roy. Soc. Open Sci. 6 191419.

-It would be insightful if the authors can present test results on the performance of the presented design in enhancing the security of the communication link and bit error rate.

-Please compare the performance of your designs with the state-of-art designs in the literature such as the ones below in order to get a better insight to the readers on the advantages of the presented work.

H. Wei, A. K. Amrithanath and S. Krishnaswamy, "3D Printing of Micro-Optic Spiral Phase Plates for the Generation of Optical Vortex Beams," in IEEE Photonics Technology Letters, vol. 31, no. 8, pp. 599-602, 15 April15, 2019.

Peter Schemmel, Giampaolo Pisano, and Bruno Maffei, "Modular spiral phase plate design for orbital angular momentum generation at millimetre wavelengths," Opt. Express 22, 14712-14726 (2014)

Review form: Reviewer 2 (Will Whittow)

Is the manuscript scientifically sound in its present form?

Yes

Are the interpretations and conclusions justified by the results?

Yes

Is the language acceptable?

Yes

Do you have any ethical concerns with this paper?

No

Have you any concerns about statistical analyses in this paper?

No

Recommendation?

Accept with minor revision (please list in comments)

Comments to the Author(s)

This is a very interesting paper. It is well written.

Please can the authors, clarify a few points to make the paper clearer for understanding.

1. I would invite the authors to re-read the abstract and conclusions and see IF they can be improved. NB i do not have any specific suggestions. My main comment is that the paper is quite complex and there will be few people who are familiar with all aspects and hence the wording of these two sections is very important.
2. All the figures are very small. I assume that this is the journal template, but if you could make them larger that would be useful.
3. Fig 1. Can the caption say what the materials and relative permittivities are.
4. Page 6. Line 47. What are the losses? Are the losses important
5. Page 6.Line 49. Why 14 wedges?
6. Can you clarify the caption to Fig 3. There is a lot of information there and the caption needs to be as clear as possible. Were measurements taken at receiving horn or extrapolated back to the SPP?
7. Fig 4 says as a function of frequency but the x axis says angle. There are lots of lines but no explanation. What is the significance of the line in bold?
8. Fig 5. As the images are incredibly small, please use different colours and more visually different symbols in each figure.
9. Fig 6 mentions the GRIN lens. Please include a photo of this and ideally a photo of it in the measurement setup.
10. Fig 7 - what is a half mirror?

Review form: Reviewer 3

Is the manuscript scientifically sound in its present form?

Yes

Are the interpretations and conclusions justified by the results?

Yes

Is the language acceptable?

Yes

Do you have any ethical concerns with this paper?

No

Have you any concerns about statistical analyses in this paper?

No

Recommendation?

Accept with minor revision (please list in comments)

Comments to the Author(s)

Comments to the authors:

Figure in [37] has offset maximum phase mode amplitude in terms of elevation angle as opposed to yours in Figure 4? Could you clarify why this is?

Some typos exist.

Can you clarify with regard to "the SPP was printed on an index-matching layer made of ABS with a lower relative permittivity of 2.65 [33] and a thickness of 5 mm" on page 5 through illustrated inclusion of this layer in Figure 1, for example?

For your conclusion, can you emphasise which SPP design would provide the best OAM system results and qualify this by saying why?

Other than these points, a good paper. Well done.

Review form: Reviewer 4

Is the manuscript scientifically sound in its present form?

No

Are the interpretations and conclusions justified by the results?

No

Is the language acceptable?

Yes

Do you have any ethical concerns with this paper?

No

Have you any concerns about statistical analyses in this paper?

No

Recommendation?

Major revision is needed (please make suggestions in comments)

Comments to the Author(s)

This is an interesting and useful comparison of three different spiral phase plate production techniques for generating OAM. It was surprising that the smoothly varying structure had notably inferior performance compared to the other two. I note that from the figure 1 photos, the smoothly varying SPP had a circular outline, while the other two were square. The stepped SPP was also used in [40], and from figure 7 in that paper we see that the rectangular SPP entirely covered the aperture of the horn. Was this the case for the three SPP's used in your paper? Could this explain the difference in performance, or was it due to some manufacturing defect in the smoothly varying SPP? Could the results of any investigation into the cause of the deficiency in the smoothly varying structure be included?

Subsection 3(e), OAM mode multiplexing, seems unclear to me. The multiplexing arrangement of figure 7 is essentially the same as figure 10 of reference [40], yet in [40] no claim was made for generating new OAM modes, it was simply a practical technique for multiplexing different OAM modes into a single composite OAM signal. In your paper we have OAM $l=1$ mode passing through the mirror, and another OAM $l=1$ mode reflected by the mirror, and due to the reflection becoming OAM $l=-1$ mode; these two orthogonal modes would be propagated to the receiver as a composite OAM signal, as described in [40]. It is not clear from your text how the OAM $l=2$ mode is generated, or how the results in figure 7 (b), (c) and (d) can be explained. I believe subsection 3(e) needs to either be re-written to explain the mechanism involved, or else deleted.

In the supplemental figures S1, S2 and S3, I presume the rows correspond to different frequencies, but no indication is given. It would be useful to indicate frequency, either by labels on the rows, or else by an indication in the captions.

Decision letter (RSOS-200493.R0)

Dear Dr Isakov,

The editors assigned to your paper ("Evaluation of the Laguerre-Gaussian Mode Purity Produced by 3D-printed Microwave Spiral Phase Plates") have now received comments from reviewers.

We would like you to revise your paper in accordance with the referee and Associate Editor suggestions which can be found below (not including confidential reports to the Editor). Please note this decision does not guarantee eventual acceptance.

Please submit a copy of your revised paper before 20-Jun-2020. Please note that the revision deadline will expire at 00.00am on this date. If we do not hear from you within this time then it will be assumed that the paper has been withdrawn. In exceptional circumstances, extensions may be possible if agreed with the Editorial Office in advance. We do not allow multiple rounds of revision so we urge you to make every effort to fully address all of the comments at this stage. If deemed necessary by the Editors, your manuscript will be sent back to one or more of the original reviewers for assessment. If the original reviewers are not available, we may invite new reviewers.

- Data accessibility

<http://datadryad.org/submit?journalID=RSOS&manu=RSOS-200493>

- Competing interests

- Authors' contributions

- Acknowledgements

- Funding statement

on behalf of Dr Derek Abbott (Associate Editor) and Miles Padgett (Subject Editor)
openscience@royalsociety.org

Reviewers' Comments to Author:

Reviewer: 1

Comments to the Author(s)

The manuscript presents the design and implementation of spiral phase plates (SPPs) using additive manufacturing for generating OAM waves. Below are my comments and questions on the presented work:

-Based on the introduction, the main contribution of the presented work is not clear. Please add explanations clarifying the main contributions behind the presented research work specially in comparison with the following article, which has been recently published by the same authors.

Allen B, Pelham T, Wu Y, Drysdale T, Isakov D, Gamlath C, Stevens C, Hilton G, Beach M, Grant PS. 2019 Experimental evaluation of 3D printed spiral phase plates for enabling an orbital angular momentum multiplexed radio system, B. Allen, T. Pelham, Y. Wu, T. Drysdale, D. Isakov, C. Gamlath, C. Stevens, G. Hilton, M. Beach and P.S Grant, Roy. Soc. Open Sci. 6 191419.

-It would be insightful if the authors can present test results on the performance of the presented design in enhancing the security of the communication link and bit error rate.

-Please compare the performance of your designs with the state-of-art designs in the literature such as the ones below in order to get a better insight to the readers on the advantages of the presented work.

H. Wei, A. K. Amrithanath and S. Krishnaswamy, "3D Printing of Micro-Optic Spiral Phase Plates for the Generation of Optical Vortex Beams," in IEEE Photonics Technology Letters, vol. 31, no. 8, pp. 599-602, 15 April15, 2019.

Peter Schemmel, Giampaolo Pisano, and Bruno Maffei, "Modular spiral phase plate design for orbital angular momentum generation at millimetre wavelengths," Opt. Express 22, 14712-14726 (2014)

Reviewer: 2

Comments to the Author(s)

This is a very interesting paper. It is well written.

Please can the authors, clarify a few points to make the paper clearer for understanding.

1. I would invite the authors to re-read the abstract and conclusions and see IF they can be improved. NB i do not have any specific suggestions. My main comment is that the paper is quite complex and there will be few people who are familiar with all aspects and hence the wording of these two sections is very important.
2. All the figures are very small. I assume that this is the journal template, but if you could make them larger that would be useful.
3. Fig 1. Can the caption say what the materials and relative permittivities are.
4. Page 6. Line 47. What are the losses? Are the losses important
5. Page 6.Line 49. Why 14 wedges?
6. Can you clarify the caption to Fig 3. There is a lot of information there and the caption needs to be as clear as possible. Were measurements taken at receiving horn or extrapolated back to the SPP?
7. Fig 4 says as a function of frequency but the x axis says angle. There are lots of lines but no explanation. What is the significance of the line in bold?
8. Fig 5. As the images are incredibly small, please use different colours and more visually different symbols in each figure.
9. Fig 6 mentions the GRIN lens. Please include a photo of this and ideally a photo of it in the measurement setup.
10. Fig 7 - what is a half mirror?

Reviewer: 3

Comments to the Author(s)

Comments to the authors:

Figure in [37] has offset maximum phase mode amplitude in terms of elevation angle as opposed to yours in Figure 4? Could you clarify why this is?

Some typos exist.

Can you clarify with regard to "the SPP was printed on an index-matching layer made of ABS with a lower relative permittivity of 2.65 [33] and a thickness of 5 mm" on page 5 through illustrated inclusion of this layer in Figure 1, for example?

For your conclusion, can you emphasise which SPP design would provide the best OAM system results and qualify this by saying why?

Other than these points, a good paper. Well done.

Reviewer: 4

Comments to the Author(s)

This is an interesting and useful comparison of three different spiral phase plate production techniques for generating OAM. It was surprising that the smoothly varying structure had notably inferior performance compared to the other two. I note that from the figure 1 photos, the smoothly varying SPP had a circular outline, while the other two were square. The stepped SPP was also used in [40], and from figure 7 in that paper we see that the rectangular SPP entirely covered the aperture of the horn. Was this the case for the three SPP's used in your paper? Could this explain the difference in performance, or was it due to some manufacturing defect in the smoothly varying SPP? Could the results of any investigation into the cause of the deficiency in the smoothly varying structure be included?

Subsection 3(e), OAM mode multiplexing, seems unclear to me. The multiplexing arrangement of figure 7 is essentially the same as figure 10 of reference [40], yet in [40] no claim was made for generating new OAM modes, it was simply a practical technique for multiplexing different OAM modes into a single composite OAM signal. In your paper we have OAM $l=1$ mode passing through the mirror, and another OAM $l=1$ mode reflected by the mirror, and due to the reflection becoming OAM $l=-1$ mode; these two orthogonal modes would be propagated to the receiver as a composite OAM signal, as described in [40]. It is not clear from your text how the OAM $l=2$ mode is generated, or how the results in figure 7 (b), (c) and (d) can be explained. I believe subsection 3(e) needs to either be re-written to explain the mechanism involved, or else deleted.

In the supplemental figures S1, S2 and S3, I presume the rows correspond to different frequencies, but no indication is given. It would be useful to indicate frequency, either by labels on the rows, or else by an indication in the captions.

Author's Response to Decision Letter for (RSOS-200493.R0)

See Appendix A.

Decision letter (RSOS-200493.R1)

Dear Dr Isakov,

It is a pleasure to accept your manuscript entitled "Evaluation of the Laguerre-Gaussian Mode Purity Produced by 3D-printed Microwave Spiral Phase Plates" in its current form for publication in Royal Society Open Science. The comments of the reviewer(s) who reviewed your manuscript are included at the foot of this letter.

on behalf of Dr Derek Abbott (Associate Editor) and Miles Padgett (Subject Editor)
openscience@royalsociety.org

Appendix A

Royal Society Open Science Manuscript ID: RSOS-200493

Authors response to the reviewers on
“Evaluation of the Laguerre-Gaussian Mode Purity
Produced by 3D-printed Microwave Spiral Phase
Plates”

D. Isakov*, Y. Wu, B. Allen, P. S. Grant, C. J. Stevens, G. Gibbons

June 24, 2020

Dear Editor,

We would like to thank the reviewers for their constructive comments on this article. We hope that this new version addresses all the queries and suggested changes of the reviewers.

Below, we reply to each of the reviewer comments and the revised manuscript shows the corresponding changes in red.

With regards,
Dr. Dmitry Isakov

*d.isakov@warwick.ac.uk

Reviewer 1

The manuscript presents the design and implementation of spiral phase plates (SPPs) using additive manufacturing for generating OAM waves. Below are my comments and questions on the presented work:

Q1.1 *Based on the introduction, the main contribution of the presented work is not clear. Please add explanations clarifying the main contributions behind the presented research work specially in comparison with the following article, which has been recently published by the same authors.*

A1.1 The previous work [40] focused on the design, fabrication and preliminary performance of different, stepwise SPPs. Only one of three SPPs in the current work is the same as that in [40]. [40] also suggested a use case for the SPPs in a communication system, but presented no experimental demonstration or data of any type. In contrast, the current paper engineers the best performing SPPs into a rudimentary communications system demonstration – it is a complementary but significant development of [40]. For the first time to our knowledge, we show a flat, not-stepped or sloped, GRIN SPP manufactured using dual-head material extrusion 3D printing. Our results show that when coupled with a lens, the GRIN SPP has superior OAM mode purity. We have re-written the Conclusions to bring this out more clearly:

“... The different SPP geometries gave different microwave beam refraction characteristics. SPPs with both free surfaces perpendicular to the incident beam direction—a SPP with an azimuthal staircase structure and a parallel sided SPP with an azimuthal gradient refraction index—tended to provide OAM modes of relatively high, and useful, purity. Surprisingly, the third SPP that had a smooth azimuthal variation in thickness of a single material showed the lowest overall extent of OAM mode purity, because (i) the “upper” (Figure 1a) non-normal free surface promoted internal reflections that contributed to interference and superposition of radial modes; and (ii) the discontinuity produced by the single large step was of thickness close to the incident wavelength and may have promoted a tendency for internal standing waves. Combining the 3D-Printed SPPs with a 3D-Printed dielectric lens produced distinct improvements in OAM mode purity and reduced OAM beam divergence.

And:

... The best overall OAM purity and reduced OAM beam divergence was provided by a flat, relatively thin SPP with an azimuthally varying

graded index combined with a dielectric lens. Subsequently, two SPPs, a dielectric lens and a dielectric half-mirror were then assembled into a rudimentary microwave multi-mode multiplexed transmission and reception system. Preliminary data successfully confirmed multi-mode and multiplexed transmission and reception for a range of SPP configurations.”

Q1.2 *It would be insightful if the authors can present test results on the performance of the presented design in enhancing the security of the communication link and bit error rate.*

A1.2 We agree this would be very illuminating and is the focus of our current work, but suggest it is a significant undertaking in its own right and out of scope of the current paper. However, we have added a sentence on page 7 to highlight future possibilities as follows:

Minimising inter-modal interference is a key parameter in the design of an OAM link. As with any radio link, sufficient signal-to-interference-plus-noise ratio would have to be achieved to restrict the bit error rate. A further particular aspect of an OAM based communication system is the possibility that it may offer for enhanced security, and this is the focus of current work.

Q1.3 *Please compare the performance of your designs with the state-of-art designs in the literature such as the ones below in order to get a better insight to the readers on the advantages of the presented work.*

1 Ref [31]

2 H. Wei, et al. IEEE Phot. Tech. Lett. 31, 599 (2019).

A1.3 [31] reported a modular split stepped $\Delta l = \pm 10$ SPP machined from polypropylene, and the numerical analyses of the “split-stepped” and smooth $l = \pm 1$ SPPs. Although our results show a similar mode content for all 3D-printed SPPs around 80% (for frequencies close to the nominal 15 GHz), our “smooth” SPP has the lowest mode purity, whereas [31] suggested the mode power for modes $l = \pm 1$ were slightly improved over the stepped SPP equivalent. We have added the following sentence on page 7 to clarify the result:

Taken together, the results in Figures 3 and 4 were in good qualitative agreement with the theoretical analysis described in [37, 38] for a vertex beam generated by an antenna array, which also showed that the energy of the fundamental mode decreased with increasing elevation

angle, whereas the energy of the harmonic modes increased. Whereas a previous comparison between SPPs with either stepped or smooth azimuthal thickness changes suggested a small advance for the smooth case [31], our results for modes $l = \pm 1$ show that stepped SPP had a small percentage improvement over the smooth SPP. However, in both cases the differences were relatively small.

The very nice work by H. Wei et al. concerned a nano-lithography fabricated optical wavelength SPP and its performance analysis using the Michelson interferometric technique. However, since no OAM mode purity analysis was given and only interference snapshots, it is not straightforward (and probably misleading) to compare between the two very different works robustly, and we suggest therefore that the comparison with [31] only (as above) is included.

Reviewer 2

This is a very interesting paper. It is well written. Please can the authors, clarify a few points to make the paper clearer for understanding.

Q2.1 *I would invite the authors to re-read the abstract and conclusions and see IF they can be improved. NB i do not have any specific suggestions. My main comment is that the paper is quite complex and there will be few people who are familiar with all aspects and hence the wording of these two sections is very important.*

A2.1 Thank you for the suggestion. We have now completely revised the Conclusions to improve clarity, and there are many other improvements of this type throughout the revised version (in red). The Conclusions now read:

“... The different SPP geometries gave different microwave beam refraction characteristics. SPPs with both free surfaces perpendicular to the incident beam direction—a SPP with an azimuthal staircase structure and a parallel sided SPP with an azimuthal gradient refraction index—tended to provide OAM modes of relatively high, and useful, purity. Surprisingly, the third SPP that had a smooth azimuthal variation in thickness of a single material showed the lowest overall extent of OAM mode purity, because (i) the “upper” (Figure 1a) non-normal free surface promoted internal reflections that contributed to interference and superposition of radial modes; and (ii) the discontinuity produced by the single large step was of thickness close to the

incident wavelength and may have promoted a tendency for internal standing waves. Combining the 3D-Printed SPPs with a 3D-Printed dielectric lens produced distinct improvements in OAM mode purity and reduced OAM beam divergence.

The best overall OAM purity and reduced OAM beam divergence was provided by a flat, relatively thin SPP with an azimuthally varying graded index combined with a dielectric lens. Subsequently, two SPPs, a dielectric lens and a dielectric half-mirror were then assembled into a rudimentary microwave multi-mode multiplexed transmission and reception system. Preliminary data successfully confirmed multi-mode and multiplexed transmission and reception for a range of SPP configurations.”

Q2.2 *All the figures are very small. I assume that this is the journal template, but if you could make them larger that would be useful.*

A2.2 All figures have now been increased in size.

Q2.3 *Fig 1. Can the caption say what the materials and relative permittivities are.*

A2.3 Done.

Q2.4 *Page 6. Line 47. What are the losses? Are the losses important*

A2.4 The information about losses has now been added on page 3 and page 4. The losses are typical for these materials, were carefully measured using dielectric split post resonator technique, and were discussed more fully in previous works [32] and [33] on materials formulation.

Q2.5 *Page 6.Line 49. Why 14 wedges?*

A2.5 Wedges are needed because it is not possible to mix the high and low permittivity printed materials on a microscopic scale, only by placing them side-by-side. As long as this discretization is done on a sub-wavelength length scale, the effective resulting permittivity of the region is weighted mixture of the two wedge permittivities. 28 alternating wedges of high and low permittivity materials was the smallest practical discretization of the azimuthal permittivity variation achievable. We have reworded the explanation in the text on page 4–5 as follows:

The flat SPP design was split into 28 wedges of variable size—14 comprised high permittivity BaTiO₃/ABS and 14 comprised low permittivity ABS. “High/low” pairs of wedges formed a 14 equal-sized segments in which the relative proportion (or in-plane area) of the different wedges changed according to azimuthal position (see Figure 1(c)) to produce a different average, effective permittivity simply given by the volume weighted proportion of the two sub-wedges. The 14 segments (total 28 wedges) were the smallest discrete segments that could be reliably printable (further increasing the wedge number and decreasing the wedge size made the required resolution in the central part of the SPP too fine-scale to be printed). By keeping the thickness of the SPP constant at ...

Q2.6 *Can you clarify the caption to Fig 3. There is a lot of information there and the caption needs to be as clear as possible. Were measurements taken at receiving horn or extrapolated back to the SPP?*

A2.6 We have clarified the figure caption as follows:

“The radiation signal passed through an SPP with a staircase design (as an example) in an X-Y plane of area $700 \times 700 \text{ mm}^2$ perpendicular to the propagation vector. The experimental arrangement is shown in Figure 2. Top row—the S_{21} amplitude profile; middle row—the S_{21} phase profile; bottom row—the OAM mode spectrum. The circle superimposed on the top row, left corner figure has its centre at the location corresponding to the minimum of the amplitude intensity. The corresponding phase data on the second row was obtained by moving through the data points on the circle circumference and at each point decomposing the amplitude data using a Fourier transform to obtain the phase profile (second row) and the OAM mode (third row).

Q2.7 *Fig 4 says as a function of frequency but the x axis says angle. There are lots of lines but no explanation. What is the significance of the line in bold?*

A2.7 The bold line is the frequency at which the mode energy ratio had the highest possible value over a relatively large area (angle). The other grey lines represent cases (frequencies) of lower mode energy ratio. The ‘purest’ OAM mode is that with highest overall the mode energy ratio over the angle range studied. We have clarified the text on page 7 as follows:

Each of the lines is the experimental data for a specific frequency, as the angle is varied over the range. In most cases, the intended mode could only be maintained at relatively small elevation angles, typically $\theta < 5^\circ$ beyond which the mode degenerated due to overlapping of multiple phase modes. The thicker black line in each figure indicates the frequency that provided the overall best performing, purest response (over the range) for each SPP.

In addition, the caption for Figure 4 on page 7 has been modified as follows:

Mode energy ratio as a function of elevation angle for the 3D-Printed SPPs. Each grey plot-line represents a different discrete frequency. (a) SPP with a smooth curvature surface; (b) SPP with a staircase structure; and (c) a flat SPP with the gradient refractive index. The bold lines in each sub-figure represents the frequency at which OAM mode l had the highest overall purity.

Q2.8 *Fig 5. As the images are incredibly small, please use different colours and more visually different symbols in each figure.*

A2.8 The images and text have been increased in size.

Q2.9 *Fig 6 mentions the GRIN lens. Please include a photo of this and ideally a photo of it in the measurement setup.*

A2.9 The lens used was a bi-convex dielectric lens, not a GRIN lens. We have clarified the experimental section on page 3 and provided the reference as follows:

A dielectric bi-convex lens was used to collimate the divergence of the OAM waves and was manufactured using a PolyJet 3D Printer (J750, Stratasys). The lens was 90 mm in diameter and had a focal length of 110 mm at 15 GHz, and further details can be found in [40].

And again on page 7:

Therefore, to reduce beam divergence and to investigate the effect of lensing on OAM mode purity, a dielectric bi-convex lens with focal length $f = 110$ mm and a diameter of 90 mm was 3D-Printed using a photocurable resin of refractive index $n = 1.73$ [40].

Q2.10 *Fig 7 - what is a half mirror?*

A2.10 When oriented at $\pi/4$ to the incident beam, a half-mirror is able to refract 50% of the beam intensity and transmit the rest 50%. We have modified the figure caption to Figure 7 to explain as follows:

“...Left insert: a schematic of the experimental setup. **When the half mirror was oriented at $\pi/4$ to the incident beam, 50% of the incident beam intensity was refracted and 50% was transmitted.** The resulting (a) S_{21} amplitude profile; (b) S_{21} phase profile; (c) OAM mode spectrum; and (d) mode energy ratio.”

Reviewer 3

Q3.1 *Figure in [37] has offset maximum phase mode amplitude in terms of elevation angle as opposed to yours in Figure 4? Could you clarify why this is?*

A3.1 The offset maximum phase mode amplitude in [37] is because the numerical simulation and OAM mode analysis in [37] is for a n -element circular array. Using this approximation for a dielectric SPP will give $N \rightarrow \infty$ and $R \rightarrow 0$ (where N is number of array elements and R is the distance of the array element from the centre), resulting in a canonical expression for total vector potential and where the phase dislocation uncertainty has infinitely small area and therefore will show no offset. Other than this difference, our results are in good agreement with the model in [37] and we have the following statement on page 7:

“Taken together, the results in Figures 3 and 4 were in good qualitative agreement with the theoretical analysis described in [37, 38] for a vertex beam generated by an antenna array, which also showed that the energy of the fundamental mode decreased with increasing elevation angle, whereas the energy of the harmonic modes increased.”

Q3.2 *Some typos exist.*

A3.2 The manuscript has been carefully revised and proof read several times for English grammar and typographic errors.

Q3.3 Can you clarify with regard to “the SPP was printed on an index-matching layer made of ABS with a lower relative permittivity of 2.65 [33] and a thickness of 5 mm” on page 5 through illustrated inclusion of this layer in Figure 1, for example?

A3.3 We have modified Figure 1(b) to label more clearly where the index matching layer (IML) was located.

Q3.4 *For your conclusion, can you emphasise which SPP design would provide the best OAM system results and qualify this by saying why?*

A3.4 On page 10 we have modified the Conclusions as follows:

... The different SPP geometries gave different microwave beam refraction characteristics. SPPs with both free surfaces perpendicular to the incident beam direction—a SPP with an azimuthal staircase structure and a parallel sided SPP with an azimuthal gradient refraction index—tended to provide OAM modes of relatively high, and useful, purity. Surprisingly, the third SPP that had a smooth azimuthal variation in thickness of a single material showed the lowest overall extent of OAM mode purity, because (i) the “upper” (Figure 1a) non-normal free surface promoted internal reflections that contributed to interference and superposition of radial modes; and (ii) the discontinuity produced by the single large step was of thickness close to the incident wavelength and may have promoted a tendency for internal standing waves. Combining the 3D-Printed SPPs with a 3D-Printed dielectric lens produced distinct improvements in OAM mode purity and reduced OAM beam divergence.

The best overall OAM purity and reduced OAM beam divergence was provided by a flat, relatively thin SPP with an azimuthally varying graded index combined with a dielectric lens. Subsequently, two SPPs, a dielectric lens and a dielectric half-mirror were then assembled into a rudimentary microwave multi-mode multiplexed transmission and reception system. Preliminary data successfully confirmed multi-mode and multiplexed transmission and reception for a range of SPP configurations.

Reviewer 4

Q4.1 *This is an interesting and useful comparison of three different spiral phase plate production techniques for generating OAM. It was surprising that the smoothly varying structure had notably inferior performance compared to the other two. I note that from the figure 1 photos, the smoothly varying SPP had a circular outline, while the other two were square. The stepped SPP was also used in [40], and from figure 7 in that paper we see that the*

rectangular SPP entirely covered the aperture of the horn. Was this the case for the three SPP's used in your paper? Could this explain the difference in performance, or was it due to some manufacturing defect in the smoothly varying SPP? Could the results of any investigation into the cause of the deficiency in the smoothly varying structure be included?

A4.1 The smooth SPP has a diameter slightly larger than the diagonal of the aperture of the horn antenna (we have added this information in Methods, page 3) so a mismatch in sizes was not the reason for the “smooth” SPP performance to be below that of the step-wise and GRIN counterparts. We have clarified that section of the Conclusions as follows:

The different SPP geometries gave different microwave beam refraction characteristics. SPPs with both free surfaces perpendicular to the incident beam direction—a SPP with an azimuthal staircase structure and a parallel sided SPP with an azimuthal gradient refraction index—tended to provide OAM modes of relatively high, and useful, purity. Surprisingly, the third SPP that had a smooth azimuthal variation in thickness of a single material showed the lowest overall extent of OAM mode purity, because (i) the “upper” (Figure 1a) non-normal free surface promoted internal reflections that contributed to interference and superposition of radial modes; and (ii) the discontinuity produced by the single large step was of thickness close to the incident wavelength and may have promoted a tendency for internal standing waves. Combining the 3D-Printed SPPs with a 3D-Printed dielectric lens produced distinct improvements in OAM mode purity and reduced OAM beam divergence.

Also on page 3:

The size of the SPPs was chosen to fully cover the aperture of the antenna horn.

Q4.2 *Subsection 3(e), OAM mode mutliplexing, seems unclear to me. The multiplexing arrangement of figure 7 is essentially the same as figure 10 of reference [40], yet in [40] no claim was made for generating new OAM modes, it was simply a practical technique for multiplexing different OAM modes into a single composite OAM signal. In your paper we have OAM $l=1$ mode passing through the mirror, and another OAM $l=1$ mode reflected by the mirror, and due to the reflection becoming OAM $l=-1$ mode; these two orthogonal modes would be propagated to the receiver as a composite OAM signal, as described in [40]. It is not clear from your text how the OAM*

l=2 mode is generated, or how the results in figure 7 (b), (c) and (d) can be explained. I believe subsection 3(e) needs to either be re-written to explain the mechanism involved, or else deleted.

A4.3 We apologise for the confusion, and we are grateful for the reviewer taking the trouble to look at our prior work in [40]. First, to clarify: the multiplexing arrangement described in [40] was only a schematic and theorised, not an experimental arrangement that was used in the paper or that we had achieved at that time; [40] contains no results from this type of set-up. The current paper reports the first experimental results from this suggested arrangement, and confirms its usefulness and performance.

Second, the reviewer is correct that the arrangement in Figure 7 must include a pair of $l = 1$ and $l = +1$ SPPs to convert these to mode $l = 2$, and is given in the text on page 8: “To obtain a summation of OAM modes, SPPs with positive and negative topology must be used simultaneously, due to the phase sign change on reflection.” We have modified the Figure 7 caption with the correct notation that now matches the text:

Multiplexing of two OAM modes with $l = 1+1$. **Note the mode $l = -1$ changed its polarity on reflection.** Left insert: . . .

Please note that the manuscript also reports on the case of a pair of SPPs with $l = 1$ that generates a plane wave. Both case are shown in Supplementary Materials, Figure S3.

Q4.3 *In the supplemental figures S1, S2 and S3, I presume the rows correspond to different frequencies, but no indication is given. It would be useful to indicate frequency, either by labels on the rows, or else by an indication in the captions.*

A2.8 Done—more informative captions have been added.